# An adaptive peptide-binding site in ubiquitin receptor hRpn13 revealed by structural studies

Bakar Hassan [1,3], Monika Chandravanshi[1,3], Martin Y. Ng [2], Hitendra Negi[1], Brice A. P. Wilson[2] & Kylie J. Walters [1] ✉

A pleckstrin-like receptor for ubiquitin (Pru) domain in hRpn13 binds ubiquitin and proteasome subunit hRpn2. Here, we report a crystal structure of Pru bound to amino acids at the extreme N-terminus (ENT) of recombinant hRpn13. ENT adopts a U shape with native sequence along one side where M1 is buried in a Pru W108-centered pocket, and non-native sequence along the other with main chain hydrogen bonding to a neighboring Pru of the crystal lattice. These ENT:Pru interactions are stable in molecular dynamics simulations even with inclusion of only one Pru. Our findings suggest that hRpn13 can form bidentate interactions with ubiquitinated substrates by binding to both ubiquitin chains and disordered sequences of substrates. Testing this model by solution nuclear magnetic resonance revealed Pru to bind weakly to various peptides, concurrent binding with ubiquitin, and ENT displacement by hRpn2, the latter required for substrate handoff to the proteasome ATPases.

The ubiquitin-proteasome system (UPS) is responsible for regulated protein degradation in eukaryotes. It begins with substrate ubiquitination by a multi-tiered enzymatic cascade and terminates with proteolysis at the 26S or 30S proteasome[1,2]. Inhibitors of the proteasome core particle (CP), where proteolysis occurs, are standard of care for patients with multiple myeloma[3].

A therapeutic window for cancer is generated by its greater dependency on the UPS from higher levels of misfolded proteins due to hypoxia[4], glucose deprivation[5], oxidative stress[6], mutations, and/or stoichiometric imbalance from aneuploidy[7].

The CP is capped by a 19S regulatory particle (RP) that houses three ubiquitin-binding substrate receptors, Rpn1, Rpn10 and Rpn13[2]. These proteins use distinct modes of substrate recognition, with Rpn1[8] and Rpn10[9,10] binding ubiquitin through helical regions and Rpn13 using an N-terminal Pru (pleckstrin-like receptor for the ubiquitin) domain[11–13]. Each substrate receptor also binds UBL (ubiquitin-like) domains within ubiquitin-binding proteasome shuttle factors[14,15] and ubiquitin processing enzymes[8,16–20]. A C-terminal DEUBAD domain in hRpn13 binds and activates the deubiquitinase UCHL5/Uch37[17–19],

which hydrolyzes branched ubiquitin chains of mixed chain linkage (K48/K6, K48/K11, and K48/K63) and not homotypic K48-linked chains[21].

hRpn13 Pru binds proteasomes at the extreme C-terminus of hRpn2, which is otherwise intrinsically disordered[22–24]. The hRpn2-binding surface of hRpn13 has emerged as a therapeutic target, as small molecule interactions at this surface induce apoptosis and prevent tumor growth in myeloma and ovarian cancer xenograft models[25–27]. Small molecule proteolysis targeting chimeras (PROTACs) that target hRpn13 were used as chemical probes to identify in multiple myeloma cells, a DEUBAD-deleted hRpn13 fragment (hRpn13$^{Pru}$) that cannot bind UCHL5 and has lost regulatory restrictions imposed by an inter-domain DEUBAD:Pru interaction[11,28]. More recently, hRpn13 has also been found to function in gene expression, with its gene editing or PROTAC targeting causing a rewiring of the proteome and transcriptome[29], and to bind histone deacetylase 8 (HDAC8)[29,30] and bone marrow/spleen-specific protein-arginine deiminase type-4 (PADI4)[29].

[1]Protein Processing Section, Center for Structural Biology, Center for Cancer Research, National Cancer Institute, National Institutes of Health, Frederick, MD, USA. [2]Molecular Targets Program, Center for Cancer Research, National Cancer Institute, Frederick, MD, USA. [3]These authors contributed equally: Bakar Hassan, Monika Chandravanshi. ✉e-mail: kylie.walters@nih.gov

Here, we report that hRpn13 Pru also acts as a generalized peptide-binding domain. This discovery was made by the serendipitous crystallization of hRpn13 with its Pru domain binding to the extreme N-terminus (ENT) of the recombinant hRpn13 protein. This region includes four non-native amino acids that form a U-shaped structure with hRpn13 native amino acids M1 – T3. We used molecular dynamics (MD) simulations to further define the molecular mechanisms of hRpn13 Pru interaction with ENT and nuclear magnetic resonance (NMR) spectroscopy to validate the interaction of Pru with peptide sequences in solution. Our findings suggest a mechanism of hRpn13 interaction with ubiquitinated substrates and are useful to the advancement of structure-based screening for hRpn13 Pru binders and, more generally, for protein engineering efforts.

## Results

### hRpn13 Pru crystallizes bound to a U-shaped peptide

We attempted to crystallize hRpn13 with small molecule ligands by microbatch-under-oil, and for comparison, also setup crystallization trays for hRpn13 without ligands present. Crystals were obtained for hRpn13 in the absence of ligands that diffracted to 1.85 Å resolution with one hRpn13 molecule per asymmetric unit. Processing the diffraction data revealed the unit cell to have different dimensions compared to the previously solved apo structure (PDB 5IRS)[14] and additional electron density at the protomer interface (Fig. 1a compared to Supplementary Fig. 1a). The overall structure of hRpn13 Pru was not altered and comparison of the two structures revealed a backbone root-mean-square deviation of 0.271 Å (Supplementary Fig. 1b, purple versus black). The additional electron density was observed at the hRpn2-binding location, with the central part overlapping with hRpn2 P944 – E946[23] (PDB 6CO4, Fig. 1b). Compared to the previously solved apo structure, differences were observed in the region spanning P89 – R92 (Supplementary Fig. 1c, black versus purple), which is part of the β6/β7 loop. The position of the additional electron density was

occupied by two dithiothreitol (DTT) molecules, one of which forms a disulfide bond with C88 (Fig. 1c) in the previous structure (PDB 5IRS)[14]. DTT is a thiol-containing reducing agent that dissociates non-native disulfide bonds and does not typically remain bound to cysteine. hRpn13 Pru has four cysteines (C60, C80, C88, and C121) and in this study, DTT was substituted by tris(2-carboxyethyl)phosphine (TCEP). We therefore concluded that the previous structure was influenced by DTT being trapped at hRpn13 Pru C88.

We next sought to identify the additional density at the hRpn2-binding surface. hRpn13 (1–150) was expressed in frame with a histidine tag and PreScission protease cleavage site for affinity chromatography to ultimately yield four non-native amino acids at the N-terminus following purification. We tested and found that the electron density fit well to the extreme N-terminal (ENT) amino acids (Fig. 1d), which we define as including the residual G-P-G-S sequence ([a]G-[b]P-[c]G-[d]S) of the purification tag and M1 – T3 of the native hRpn13 sequence. ENT forms a U-shaped structure with the non-native sequence along one side and the native sequence along the other (Fig. 1d, yellow and orange, respectively). ENT occupies only a relatively small fraction of the hRpn2-binding surface but binds against the β6/β7 loop spanning R84 – R92 (PDB 6CO4, Fig. 1e). This positioning causes the β6/β7 loop to form a similar configuration to hRpn2-bound hRpn13 (Fig. 1b, black versus purple), which is distinct from the previously solved apo hRpn13 DTT-bound structure (Fig. 1c, black versus purple). In summary, hRpn13 Pru can crystallize with a peptide sequence bound to the native proteasome-binding site (Fig. 1f and Supplementary Fig. 1d).

### ENT contacts two distinct Pru surfaces

ENT forms hydrophobic contacts with hRpn13 Pru M1 at the valley of the U structure surrounded by P40, S90 and W108 (Fig. 2a). These amino acids are similarly involved in proteasome binding by interacting with hRpn2 P945 and E946 (Fig. 2b). Non-native [d]S interacts with

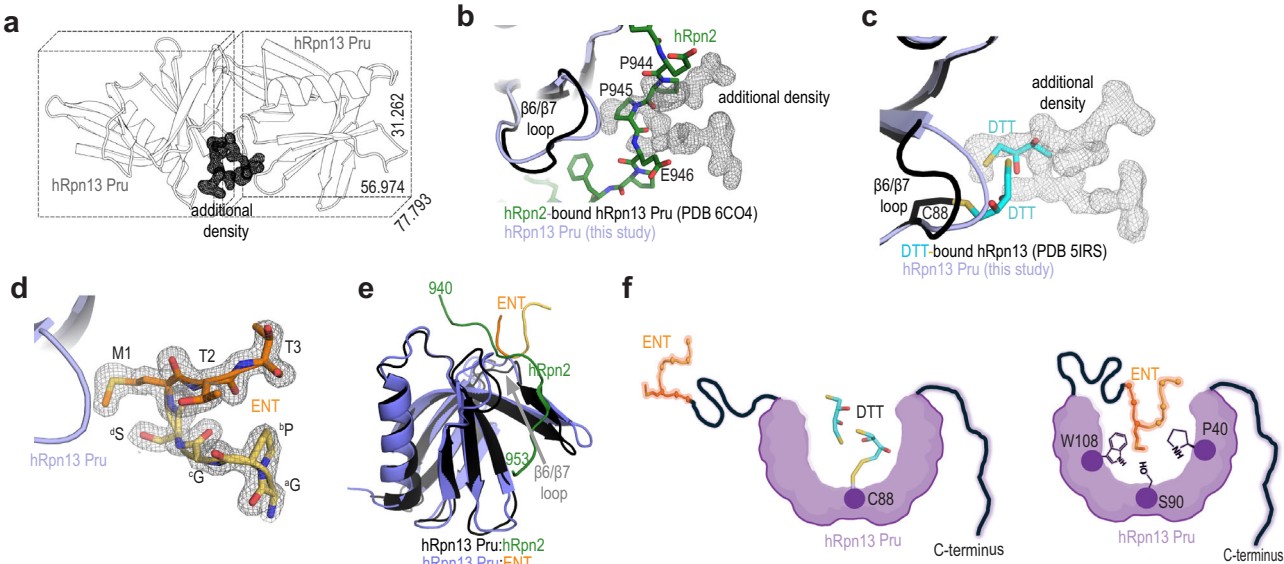

**Fig. 1 | hRpn13 Pru crystallizes with a U-shaped peptide in its proteasome-binding site. a** Ribbon diagram of neighboring hRpn13 Pru molecules within the asymmetric unit cell displayed as a black box with the unit cell dimensions denoted. Additional density between the two molecules is displayed as a black mesh. **b** Comparison of the structure of hRpn2 (green with oxygen and nitrogen in red and indigo, respectively)-bound hRpn13 (gray with the β6-β7 loop in black, PDB 6CO4) and apo hRpn13 from this study (purple) showing the additional density (gray mesh) at the hRpn2 binding surface. **c** Superimposition of apo hRpn13 Pru structures from a previous study (colored as in panel **b**) with DTT present (5IRS) and from this study (purple). DTT (cyan with oxygen and sulfur in red and yellow,

respectively) is bound to C88 and overlaps with the additional density (gray mesh) observed in this study. **d** Enlarged region of the hRpn13:ENT structure with hRpn13 displayed as a purple cartoon and stick rendering for ENT with each amino acid labeled within the *2Fo-Fc* electron density map contoured at 1.6 σ (gray mesh). The native and non-native ENT amino acids are displayed in orange and yellow, respectively. **e** Structural overlay with hRpn13 Pru superimposed for its complex with hRpn2 (green with hRpn13 in black, PDB 6CO4) or ENT (colored as in **d**) with the β6-β7 loop indicated. **f** Cartoon of DTT bound to hRpn13 Pru C88 with the terminal ends disordered (left) or of hRpn13 Pru interaction with the N-terminal ENT residues (right).

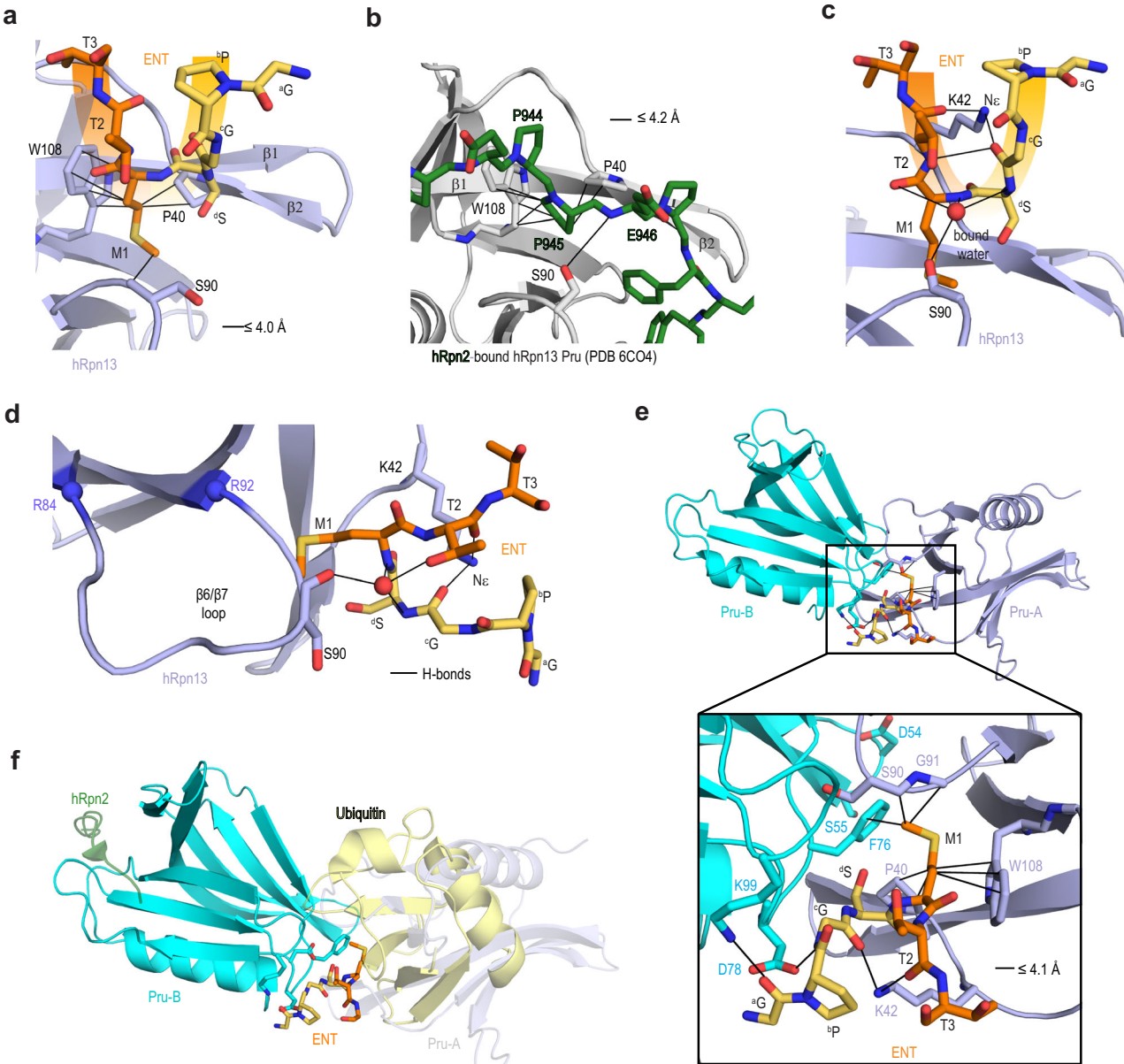

**Fig. 2 | ENT engages three hRpn13 Pru protomers of the crystal lattice.**
**a**, **b** Structural comparison of (**a**) ENT and (**b**) hRpn2 interactions with hRpn13 P40, S90 and W108. Coloring follows that of Fig. 1. **c**, **d** Expanded view of hRpn13 Pru complexed with ENT as in panel (**a**) but with display of an oxygen from a bound water molecule (red and labeled) with hydrogen bonds indicated (black line) and hydrogen bonds involving K42 Nε. A hydrogen bond between native T2 and non-native <sup>c</sup>G is also indicated. The image in panel (**d**) is rotated by 90° relative to that in panel (**c**). **e**, **f** Structure of ENT bound at the interface of three Pru molecules (labeled Pru-A, Pru-B and Pru-C) in the crystal lattice (Pru-C omitted for clarity). An expanded view in panel **e** highlights interactions between ENT and Pru-A (purple) and Pru-B (blue). Ubiquitin (yellow) and hRpn2 (green) are included in panel (**f**) by overlay of PDB 6OI4 with Pru-C hidden for clarity. The ENT molecules from the Pru-B and Pru-C asymmetric units are also hidden.

P40 of β2 (Fig. 2a) but in contrast to hRpn2 (Fig. 2b), without extending between S90 and the β1-β2 hairpin. This difference causes slight alteration in the relative positioning of the β1-β2 hairpin (Fig. 2a compared to 2b). The ENT U structure allows the K42 sidechain ε-amino group to interact with the <sup>c</sup>G and T2 backbone carbonyl oxygens and hydrogen bonding between the backbone <sup>c</sup>G carbonyl and T2 amide group (Fig. 2c).

Bound water molecules also promote the U structure and the Pru interaction with ENT. At the U hinge, a bound water bridges native and non-native amino acids by forming hydrogen bonds with the <sup>d</sup>S and M1 backbone amide groups and sidechain oxygen of T2 (Fig. 2c, d). This water also forms a hydrogen bond with the S90 backbone carbonyl oxygen (Fig. 2c, d), akin to S90 interaction with hRpn2 E946 (PDB

6CO4 Fig. 2b). Moreover, three water molecules at the other side of M1 interact with the R92 guanidinium and backbone nitrogen atoms and M109 oxygen (Supplementary Fig. 1e). Another water bridges P40 to the <sup>d</sup>S carbonyl oxygen atoms (Supplementary Fig. 1f).

We hypothesized that the interaction between ENT and Pru is supported by crystal packing, since ENT is present at the molecular interface (Fig. 1a). Analysis of ENT in the context of the neighboring asymmetric unit supports this hypothesis, as ENT M1 interacts with the W108-containing surface (as described above) of one hRpn13 molecule, named Pru-A, and the F76-containing surface of the neighboring molecule, named Pru-B (Fig. 2e). The sidechain oxygen and nitrogen atoms of Pru-B D78 and K99 form hydrogen bonds with the main chain <sup>c</sup>G amide nitrogen and <sup>a</sup>G carbonyl oxygen, respectively (Fig. 2e,

insert). The ENT-binding surface of Pru-B interacts with ubiquitin (Fig. 2f), suggesting that hRpn13 Pru can use either its hRpn2-binding or ubiquitin-binding surfaces to interact weakly with peptide sequences.

## ENT U shape and packing between Pru molecules stable by MD

We tested whether hRpn13 Pru interaction with ENT is retained over 100 ns of MD simulations. We reasoned that ENT might remain bound between Pru-A and Pru-B but lose its interaction with a neighboring molecule (Pru-C) to which it has limited contact (Supplementary Fig. 1g). We therefore used a complex containing Pru-A, Pru-B, Pru-C and ENT as a starting structure for MD (Fig. 3a, gray). Following the simulations, ENT maintained its U shape and position between Pru-A and Pru-B, whereas Pru-C shifted considerably (Fig. 3a). Root-mean-square deviations (RMSD) relative to the starting frame of Pru-A throughout the 100 ns of simulations indicated values > 20 Å for Pru-C, whereas Pru-B and ENT remained below 7 Å (Supplementary Fig. 2a). The root-mean-square fluctuations (RMSF) relative to the first frame for each Pru molecule also showed greater variance for Pru-C (Supplementary Fig. 2b).

To further evaluate the stability of ENT interaction with Pru-A and Pru-B, we plotted the distance over time between ENT M1 Cα or T2 carbonyl oxygen and Pru-A P40 Cβ or K42 Nζ, respectively, and between the ENT ${}^a$G carbonyl oxygen and Pru-B D78 Cγ or K99 Nζ (Fig. 3b). These plots indicate little variation throughout the 100 ns simulation (Fig. 3c). We also evaluated the stability of ENT interaction with Pru-C by plotting the distance the ${}^a$G amide nitrogen and Pru-C N68 Oδ1 (Fig. 3b). This distance increased within 10 ns and remained variable throughout the MD run (Fig. 3c), consistent with the shifted position of Pru-C at the end of the run (Fig. 3a).

To visualize the variability by another metric, we plotted the averaged and distances sampled at 100 picosecond intervals throughout the MD run for the aforementioned atom pairs. A similar depiction emerged whereby ENT interacted stably with Pru-A and Pru-B (Fig. 3d). Little deviation was particularly indicated for the distance between Pru-A P40 and M1 (Fig. 3d). By contrast, variance was observed for Pru-C, with a large deviation for the distance between ${}^a$G and Pru-C N68 (Fig. 3d).

To test further for Pru:ENT stability, we extended the 100 ns simulations by adding another 200 ns of MD in three replicates. In all three cases, the distances between Pru-A:ENT and Pru-B:ENT were maintained (Supplementary Fig. 3a) and ENT was retained between Pru-A and Pru-B (Supplementary Fig. 3b). Pru-C continued to shift relative to ENT with an average distance between N68 Oδ1 and the ${}^a$G amide nitrogen of > 20 Å in two of the three runs (Supplementary Fig. 3a); the distance between these atoms in the x-ray structure is 4.2 Å (Fig. 3b). Evaluating the RMSD values throughout the extended MD run relative to the last frame of Pru-A following the initial 100 ns of MD indicated large RMSD values for Pru-C (Supplementary Fig. 4a). In one of the MD runs however, Pru-C moved back towards its position in the x-ray structure (Supplementary Fig. 3b). The MD simulations did not yield any new stable interactions that persisted throughout the 200 ns in the three runs for the trimeric Pru:ENT complex. Per residue RMSF values for each of the Pru molecules in the complex were generally low, but peaked at T36, D54-S55, C88-S90, and A100 (Supplementary Fig. 4b); these amino acids localize to the center of loops (Supplementary Fig. 5a). Fluctuations at S90 are reduced for Pru-A compared to unoccupied Pru-B and Pru-C (Supplementary Fig. 4b). Correspondingly, two runs show reduced motions around the Pru-B-specific ENT-binding site for Pru-B compared to unoccupied Pru-A and Pru-C (A100, Supplementary Fig. 4b). Altogether, the MD simulations indicate ENT interaction with Pru-A and Pru-B to be stable.

## ENT stays bound but loses shape in single molecule MD runs

We tested whether the Pru-A or Pru-B interaction with ENT is by itself stable in MD calculations. The Pru-A:ENT and Pru-B:ENT complexes were isolated from the crystal structure and used as starting structures for 100 ns of MD. ENT remained bound to Pru-A and Pru-B, but lost its U shape in both cases (Fig. 4a), yielding averaged RMSD values > 8 Å for ENT in both complexes throughout the 100 ns MD simulation (Supplementary Fig. 5b, c). As in the trimer complex (Supplementary Fig. 4b), Pru-A and Pru-B per-residue RMSF values were generally low, but peaked at T36, D54, S90, and A100 (Supplementary Fig. 5d). Moreover, fluctuations near the Pru-A (S90) and Pru-B (A100) ENT-binding sites were lower for Pru-A and Pru-B, respectively, reflecting less motion by ENT occupancy. A plot of the distance over time between ${}^b$P Cβ and T3 Cα indicates loss of the U structure as early as 2 ns for Pru-A (Fig. 4b, purple) and 10 ns for Pru-B (Fig. 4b, blue). By contrast, the distance between ENT ${}^b$P Cβ and T3 Cα was maintained throughout the 100 ns MD run when three Pru molecules are included (Fig. 4b, orange), reflecting retention of the U shape (Fig. 3a).

In the initial 100 ns MD simulation with Pru-A, M1 continues to retain its interaction with P40, S90 and W108 despite loss of the ENT U structure (Fig. 4c). We plotted the change in distance over time between M1 Cγ, S, or Cβ and Pru-A P40 Cβ, S90 Oγ, or W108 Cε3, respectively, and between ${}^d$S Cβ and Pru-A P40 Cβ (Fig. 4d). The M1 interactions were stable throughout the 100 ns simulation whereas fluctuations were observed in the distance between ${}^d$S Cβ and P40 Cβ (Fig. 4e, green). These effects are also apparent in violin plots showing the averaged and sampled distances for these atom pairs (Fig. 4f, green). Distances from Pru-A P40, S90, and W108 to M1 are stable out to 200 ns for two of three independent MD runs (Supplementary Fig. 6a) despite averaged RMSD values for ENT extending above 10 Å (Supplementary Fig. 7a). For the third MD run, the averaged RMSD values are > 40 Å for ENT after ~90 ns, indicating loss of interaction.

Throughout the first 100 ns of MD, the non-native amino acids in ENT remained docked against Pru-B with retained hydrogen bonding between Pru-B D78 and K99 and ENT ${}^a$G and ${}^c$G, respectively (Fig. 4g). We quantified the changes in atom-to-atom distance throughout the initial 100 ns of MD for K99 Nζ to the carbonyl oxygen of ${}^a$G or ${}^b$P, D78 Cγ to the amide nitrogen of ${}^a$G or ${}^c$G, D79 Cγ to the amide nitrogen of ${}^d$S, or F76 Cε2 to M1 Cε (Fig. 4h). Consistent with the final structure (Fig. 4g), M1 and F76 move apart in the MD simulations (Fig. 4i, orange). An increased distance was observed in the early part of the MD run for the atom pairs involving D78 and K99, but overall, these distances remained stable, as was the distance between D79 and ${}^d$S (Fig. 4i). This effect is also apparent in violin plots of the averaged and sampled distances (Fig. 4j). These Pru-B:ENT distances are maintained out to 200 ns for two of three independent MD runs (Supplementary Fig. 6b), but ENT reached averaged RMSD values of > 30 Å for the third run (Supplementary Fig. 7a). This run also exhibited larger Pru RMSF values at D54 (Supplementary Fig. 7b). Overall, the MD runs provide support for low affinity ENT interaction at both the Pru-A and Pru-B binding sites but suggest that these interactions are stabilized by the crowded crystal lattice.

## hRpn13 Pru binds weakly to ENT by NMR

Although DTT is trapped bound to C88 in the crystal, we expected that its interaction with hRpn13 in solution is likely to be transient and dynamic. Nonetheless, we tested whether NMR spectroscopy could detect differences between hRpn13 Pru dissolved in buffer containing DTT versus TCEP. We used previously obtained NMR resonance assignments[13] and 2D ${}^1$H-${}^{15}$N HSQC experiments to detect the hRpn13 amide signals, which are sensitive to changes in their chemical environment[31], for ${}^{15}$N-labeled hRpn13 Pru with either 1 mM DTT or TCEP present in the buffer. This comparison revealed slight signal shifting, including for C88 (Supplementary Fig. 8a, b). C60 and C80 by contrast show no spectral perturbations (Supplementary Fig. 8a). Shifting is observed for C121 (Supplementary Fig. 8a, b) and mapping the amino acids sensitive to DTT versus TCEP onto the DTT-bound hRpn13 structure reveals changes that extend from C88 to the Pru

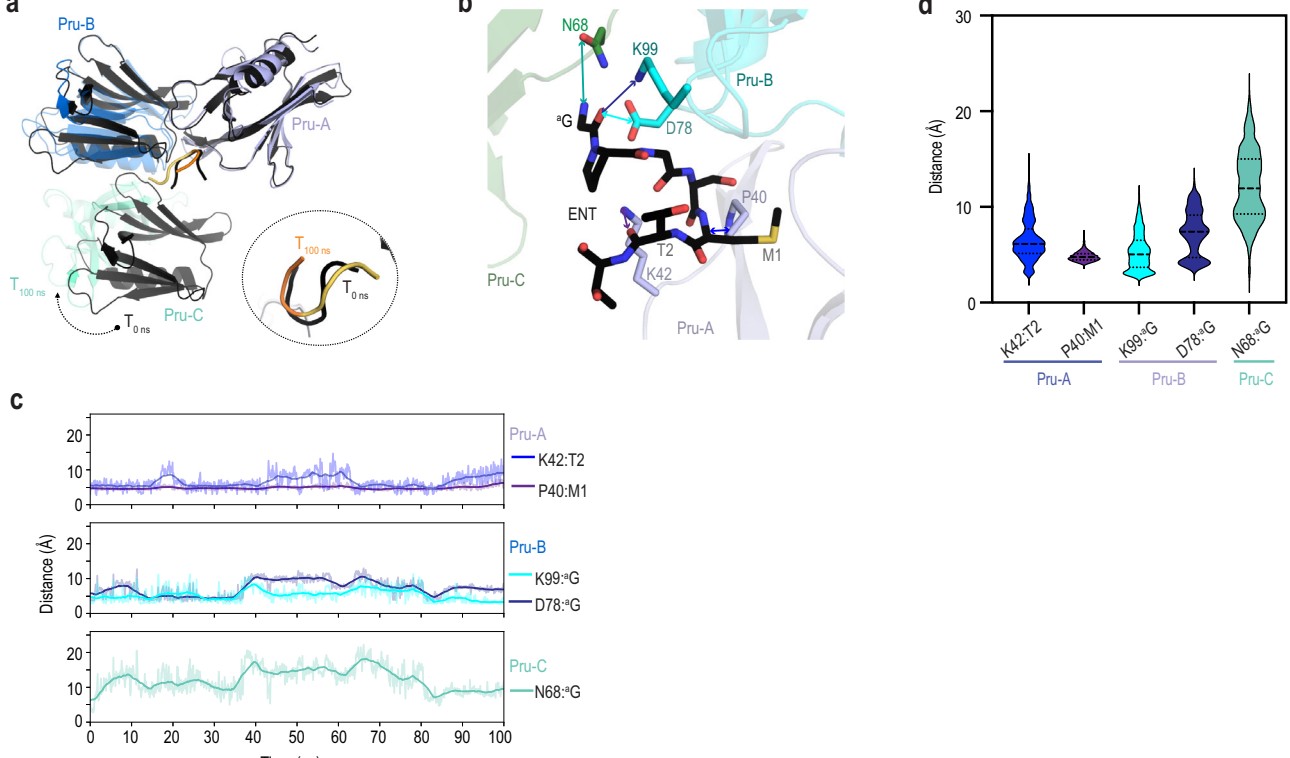

**Fig. 3 | The ENT position and U shape are retained over 100 ns of MD. a** Overlay of the ENT:Pru trimer complex before ($T_{0\,ns}$, shades of gray) and after ($T_{100\,ns}$, colored as in Fig. 2d) 100 ns of MD simulations. **b** Expanded view of the complex in (**a**), highlighting the atom pairs with distances quantified in (**c, d**). These distances are between M1 Cα and Pru-A P40 Cβ, T2 carbonyl oxygen and Pru-A K42 Nζ, ªG carbonyl oxygen with Pru-B D78 Cγ or K99 Nζ, and ªG amide nitrogen and Pru-C

N68 Oδ1. **c** Plot for ENT interactions involving Pru-A (top), Pru-B (middle), and Pru-C (bottom). **d** Violin plot of distances between atom pairs in (**b**) over 100 ns of MD ($n = 1001$). Median value is represented by a dashed line, and the first/third quartiles are represented by dotted lines. Source data for (**c, d**) are provided in the Source Data file.

helix (Supplementary Fig. 8c). It is formally possible that DTT is trapped at C121 and C88 in solution; however, the Pru helix was previously found to be allosterically sensitive to covalent binding of ligands **XL5**[28] and **XL44**[32] at C88.

We reasoned that if ENT interacts with hRpn13 Pru in solution, then hRpn2 (940 – 953), which binds hRpn13 Pru with 27 nM affinity[23], could displace it based on their common binding surface (Fig. 5a). Comparison of ${}^{1}$H-${}^{15}$N HSQC spectra of 40 µM ${}^{15}$N-labeled hRpn13 Pru without or with equimolar hRpn2 (940-953) present in TCEP-containing buffer yielded large spectral changes, consistent with hRpn2 binding. For example, V38, C88, R92, W108 and E111, which span the hRpn2 binding surface (Supplementary Fig. 1d), are affected by hRpn2 addition (Fig. 5b). In addition, signals from helical amino acids are also shifted by hRpn2 (Fig. 5b, highlighted with blue labeling), consistent with allosteric sensing at the helix. Moreover, the ${}^{d}$S signal disappeared following hRpn2 addition (Fig. 5b), but the M1 amide signal was unaffected (Fig. 5b, insert).

To test more directly whether Pru binds ENT, we generated an hRpn13 Pru variant with ENT M1 – T3, ªG, and ${}^{b}$P deleted (hRpn13_ΔMTT) for comparison with hRpn13 Pru by 2D NMR. ${}^{1}$H-${}^{15}$N HSQC spectra recorded on ${}^{15}$N-labeled hRpn13 (1–150) or hRpn13_ΔMTT indicated changes for a subset of amino acids in the Pru helix and proteasome-binding surface (Supplementary Fig. 9a). We next added a peptide encoding the ENT sequence (GPGSMTT) to 100 µM ${}^{15}$N-labeled hRpn13_ΔMTT at varying molar ratios to find small concentration-dependent shifting of the same (T114, D117-E119, C121) and additional (W61, M109, H120, H145) hRpn13 amide signals (Supplementary Fig. 9a, b), indicating weak interaction between ENT and hRpn13 Pru. We also tested and found similar spectral changes when

ENT M1 is replaced with serine by adding a peptide of sequence GPGSSTT (ENT_M1S) to 100 µM ${}^{15}$N-labeled hRpn13_ΔMTT (Fig. 5c, red versus black, and Supplementary Fig. 10a, red versus black). The similar spectral effects by adding ENT or ENT_M1S is readily apparent by quantifying and plotting the shifting across the Pru amino acid sequence (Supplementary Fig. 10b, c). We therefore concluded that hRpn13 Pru binds weakly to the ENT peptide sequence and, moreover, that this interaction does not require M1.

### Allosteric network bridges hRpn2 site to Pru helix

Painting the hRpn13 Pru amino acids that shift following ENT or ENT_M1S addition with a blue gradient color scheme onto the Pru structure bound to hRpn2 and ubiquitin[24] indicates the spectral changes to extend from the hRpn2-binding region to the helix (Supplementary Fig. 10d, e). We also mapped the affected amino acids onto a Pru structure with ENT at both the Pru-A and Pru-B positions (Fig. 5d). No effects were observed at the Pru-B-binding surface; for example, the amide signals for F76 and D78 are not among those affected (Supplementary Fig. 9b). Some amino acids near the Pru-A interaction surface shift, including R92 Hε, whereas others, such as the W108 Hε1 signal, do not (Supplementary Fig. 9b, c). The greatest spectral changes are observed where hRpn13 binds to the N-terminal end of hRpn2 (940-953, Supplementary Fig. 10d), which is proximal to where Pru-A binds ENT (Fig. 5d). This finding indicates that in solution, ENT likely binds Pru differently than in the packed crystal lattice, as also predicted by MD (Fig. 4a).

Mapping the ENT-binding site precisely is complicated by the phenomenon that the greatest effects observed in ${}^{1}$H-${}^{15}$N HSQC spectra can be caused by allosteric changes[33]. Since switching from DTT to

TCEP (Supplementary Fig. 8) or adding hRpn2 (Fig. 5b), **XL5**[28], **XL44**[32], ENT (Fig. 5c, d), or ENT_M1S (Fig. 5c) each cause signals from the Pru helix to shift, we inspected the crystal structure for a structural rationale. This effort highlighted an interaction network that connects C88 and the larger hRpn2-binding channel to the helix. The C88 backbone

oxygen atom forms water-mediated hydrogen bonds to the G91 oxygen and V93 nitrogen atoms, as the neighboring R92 side chain forms hydrogen bonds with loop residue E111 and helical residue D117 (Fig. 5e). This hydrogen-bond network allows changes at the hRpn2-binding site (Fig. 5f, shaded purple) to be allosterically sensed down

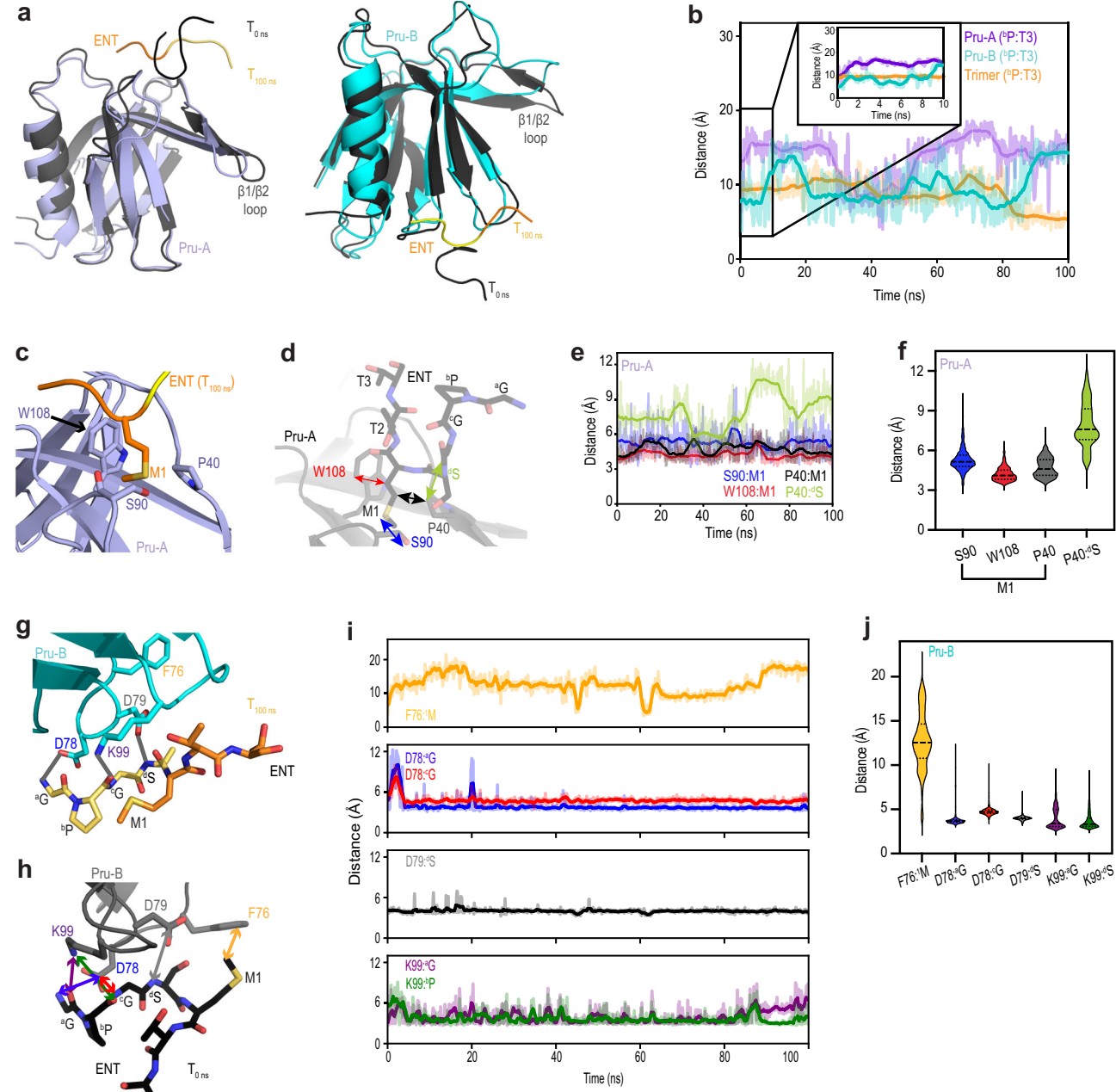

**Fig. 4 | With only one hRpn13, ENT remains bound to Pru-A or Pru-B following MD but loses its U-shape. a** Positioning of ENT in the crystal structure (0 ns, $T_{0 ns}$) and after 100 ns ($T_{100 ns}$) of MD simulation for Pru-A (purple) or Pru-B (blue). Cartoon representations are displayed with coloring of Fig. 2a for $T_{100 ns}$ and with hRpn13 and ENT in gray for ($T_{0 ns}$) starting crystal structure. **b** Plot for the distance between the Cβ atom of non-native ᵇP and Cα atom of native T3 throughout a 100 ns MD simulation performed with Pru-A (purple), Pru-B (blue) or all three (orange) hRpn13 Pru molecules. **c)** An expanded view of Pru-A with ENT (colored as in **a**) after 100 ns of MD simulation highlights interactions between M1 and P40, S90, and W108. **d** Enlarged view of hRpn13 Pru bound to ENT to highlight the atom-to-atom distances plotted in panels (**e**, **f**). These distances are between hRpn13 Pru-A M1 Cγ, S, or Cβ and P40 Cβ, S90 Oγ, or W108 Cε3, respectively and between P40 Cβ and ᵈS Cβ☐ **e, f)** Plots (n = 1001) of Pru-A:ENT pairwise distances as a function of

time (**e**) or violin plot (**f**) throughout 100 ns of molecular dynamics for the atom-atom pairs shown in panel (**d**). The median value is represented by a dashed line, and the first/third quartiles are represented by dotted lines. **g** An expanded view of Pru-B with ENT after 100 ns of MD simulation highlights interactions between D78, D79 and K99 with ᵃG, ᶜG, and ᵈS, respectively. **h** Enlarged view of hRpn13 Pru-B bound to ENT to highlight the atom-to-atom distances plotted in panels (**i, j**). These distances are between hRpn13 Pru-B F76 Cε2 to M1 Cε, D78 Cγ to ᵃG or ᶜG backbone nitrogen, D79 Cγ to ᵈS backbone nitrogen, and K99 sidechain Nζ to carbonyl of ᵃG or ᵇP. **i, j** Plots (n = 1001) of Pru-B:ENT pairwise distances as a function of time (**i**) and violin plots (**j**) throughout 100 ns of molecular dynamics for the atom-atom pairs shown in panel (**h**). Median values are represented by dashed lines, and the first/third quartiles are represented by dotted lines. Source data for (**b, e, f, i** and **j**) are provided in the Source Data file.

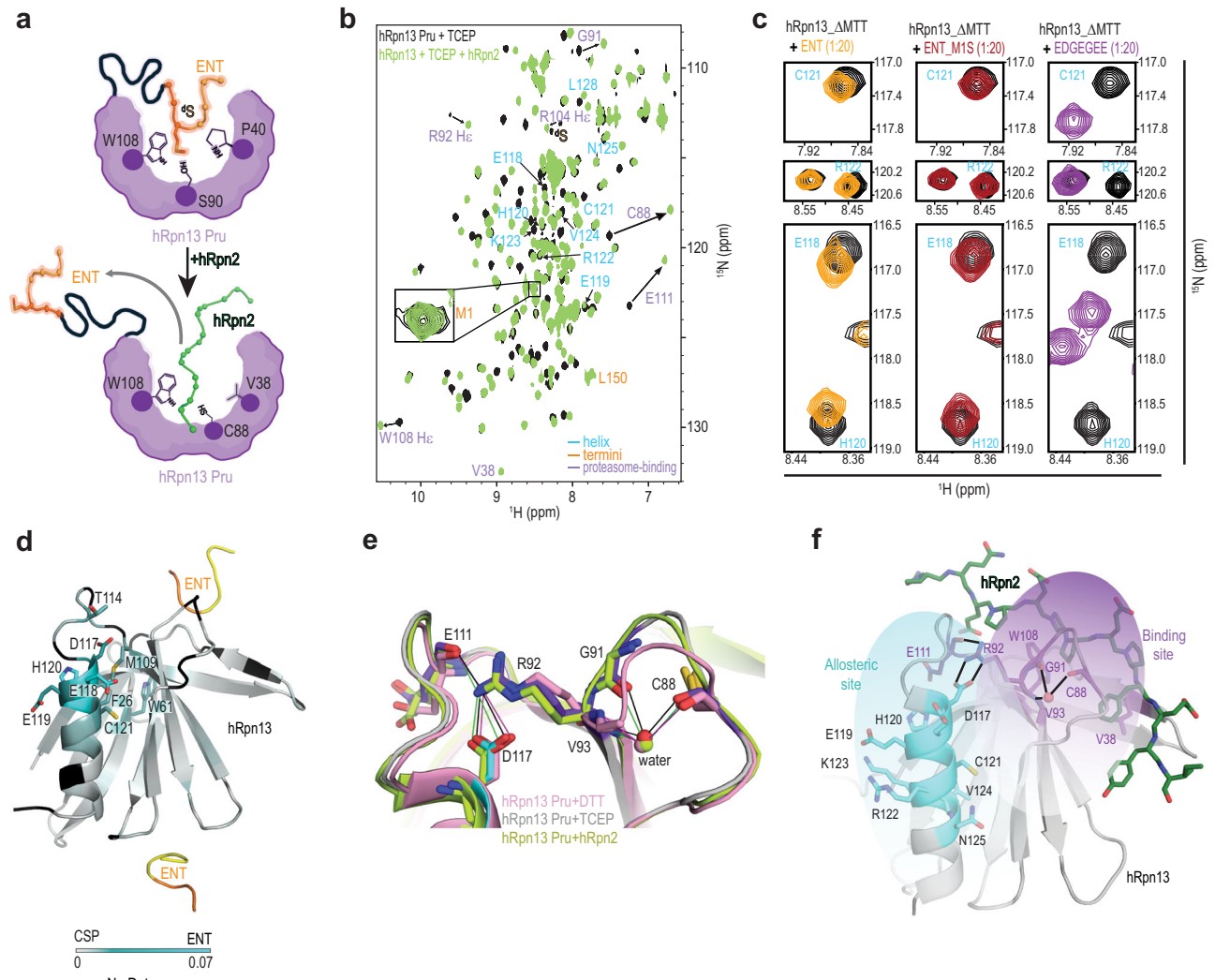

**Fig. 5 | hRpn13 Pru binds peptide sequences weakly and promiscuously.**
**a** Cartoon of ENT bound to hRpn13 Pru with M1 interacting with W108, S90, and P40 (top) and its displacement by hRpn2 (bottom). **b** [1]H-[15]N HSQC spectra of 40 µM [15]N-labeled hRpn13 Pru with 1 mM TCEP in NMR buffer without (black) and with (green) equimolar unlabeled hRpn2 (940–953). An expanded region is included to show the amide signal for ENT M1. Signals from select amino acids in the hRpn2-binding region (purple) or in the helix (blue) are labeled. The spectra were collected at 25 °C and 850 MHz with a cryogenically cooled probe. **c** Selected regions showing signals from the Pru helix of [1]H-[15]N HSQC spectra acquired on samples of 0.1 mM [15]N-labeled hRpn13_ΔMTT alone (black) and with 20-fold molar excess unlabeled ENT (orange), ENT_M1S (red), or EDGEGEE (purple). The spectra were acquired on a 600 MHz spectrometer equipped with a cryogenically cooled probe and at 25 °C in NMR buffer. **d** Ribbon representation of hRpn13 Pru-A bound to ENT, colored according to spectral changes that occur following 20-fold molar

excess addition of ENT as indicated in the figure legend. Residues with backbone amide signals that shift by greater than one standard deviation above average are displayed by a stick model, labeled. Prolines and residues whose assignments were missing or could not be determined are indicated in black. **e** Expanded view of overlayed structures of hRpn13 bound to DTT (pink), ENT (purple), or hRpn2 (green) to highlight a conserved hydrogen bond network. The oxygen of a bound water molecule is included and labeled. In hRpn13 Pru, oxygen, nitrogen, and sulfur are red, indigo, and yellow, respectively. **f** Structure of hRpn13 Pru bound to hRpn2 displayed as in Fig. 1b, showing hRpn13 sidechain atoms as sticks for amino acids with amides that shift following hRpn2 addition. The hRpn2 binding site is highlighted by purple coloring, whereas shifting in the helix is indicated by blue coloring. A hydrogen bond network is depicted with solid black lines, and the oxygen of a bound water molecule shown. Oxygen, nitrogen, and sulfur are red, indigo, and yellow, respectively.

the helix (Fig. 5f, shaded blue). A comparative analysis of Pru bound to DTT (PDB 5IRS), ENT (this study) and Rpn2 (PDB 5V1Y) illustrates that this allosteric network is preserved in all cases (Fig. 5e).

### hRpn13 Pru is highly adaptive for weak peptide binding
The similarity of spectral changes induced by ENT and ENT_M1S addition suggests that hRpn13 Pru binds peptides with plasticity, albeit the magnitude of change indicates these interactions to be weak. We reasoned that lost van der Waals contacts to M1 might be compensated by electrostatic interactions with the serine hydroxyl group since bound water bridges ENT to hRpn13 Pru in the crystal (Fig. 2c). Moreover, hRpn2 acidic residues within [940]QEPEPPEPFEYIDD[953] play

prominent roles in hRpn13 binding. We therefore tested whether an acidic peptide of sequence EDGEGEE binds to hRpn13_ΔMTT.

[1]H-[15]N HSQC spectra acquired on 100 µM [15]N-labeled hRpn13_ΔMTT with 20-fold molar excess EDGEGEE peptide displayed a greater degree of signal shifting compared to ENT (Fig. 5c, purple and Supplementary Fig. 11a) but with similar amino acids affected (Supplementary Fig. 11b compared to Supplementary Fig. 9b). This effect is readily apparent in a 2D plot of chemical shift changes for ENT and EDGEGEE (Fig. 6a) and by mapping the affected amino acids onto the hRpn13 Pru structure (Fig. 5d compared to Supplementary Fig. 11c). In addition to the Pru domain, C-terminal amino acid H145 shifts following addition of ENT (Supplementary Fig. 9b), ENT_M1S

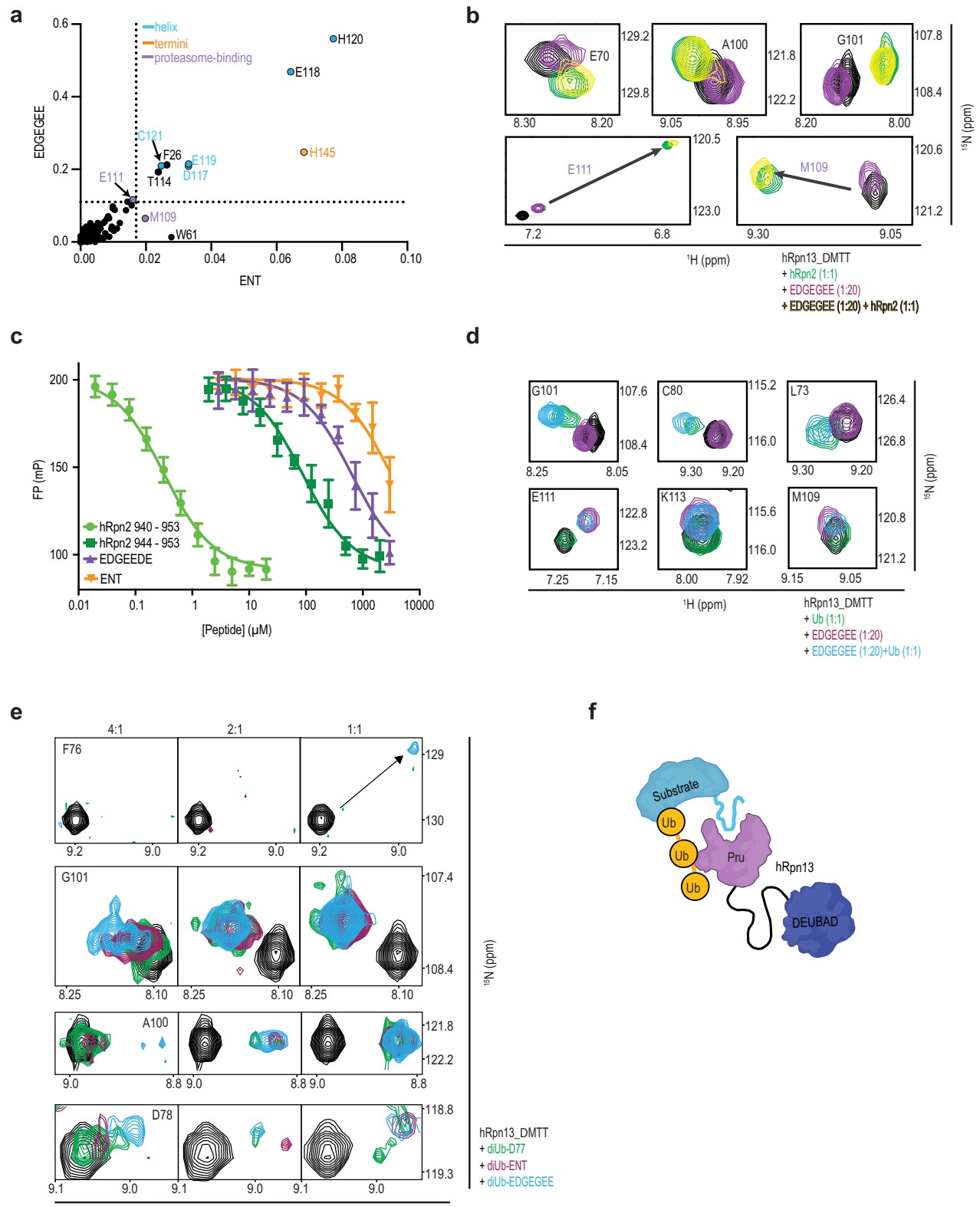

(Supplementary Fig. 10a), or EDGEGEE (Supplementary Fig. 11b). This shifting could occur by either direct interactions involving H145 or indirectly by displacement or adjustment of the C-terminal residues following Pru interaction with the added peptides.

We next tested whether hRpn2-bound hRpn13 Pru can bind to EDGEGEE by comparing $^1$H-$^{15}$N HSQC spectra of 100 μM $^{15}$N-labeled hRpn13_ΔMTT alone or with equimolar hRpn2 (940-953), 20-fold

molar excess EDGEGEE, or both components present. This comparison revealed hRpn2-specific effects when EDGEGEE and hRpn2 are both mixed with hRpn13 (Supplementary Fig. 12a), including for G29, A100 and G101 (Fig. 6b and Supplementary Fig. 12a), with additional slight shifting for some signals compared to when only hRpn2 is present (Supplementary Fig. 12a), such as E70, E111, and M109 (Fig. 6b). We also observed preservation of EDGEGEE-specific shifting for H145 and E146

**Fig. 6 | Peptide sequences compete with hRpn2 for hRpn13 but bind simultaneously with ubiquitin to allow bidentate interactions with ubiquitinated substrates. a** Plot of the signal shifting for hRpn13_ΔMTT following the addition of 20-fold molar excess ENT or EDGEGEE. Dotted lines indicate one standard deviation above average and signals from the helix, terminal ends, or proteasome-binding site are colored blue, orange, and purple, respectively. **b** Expanded regions of overlayed ¹H-¹⁵N HSQC spectra acquired on 0.1 mM ¹⁵N-labeled hRpn13_ΔMTT alone (black) or with equimolar unlabeled hRpn2 (green), 20-fold molar excess EDGEGEE (purple), or equimolar hRpn2 and 20-fold molar excess EDGEGEE (yellow). All spectra in this panel and (**d**) and (**e**) were acquired on a 600 MHz spectrometer equipped with a cryogenically cooled probe and at 25 °C. **c** Fluorescence polarization recorded during titration of unlabeled ENT (orange), EDGEGEE (purple), hRpn2 (940–953,

light green) or hRpn2 (944–953, dark green) into 10 nM of TAMRA-labeled hRpn2 (940–953) in the presence of 80 nM hRpn13. Points are reported as mean ± SD, $N = 6$ (assay replicates). **d** Expanded regions of overlayed ¹H-¹⁵N HSQC spectra acquired on samples of 0.1 mM ¹⁵N-labeled hRpn13_ΔMTT alone (black) or with equimolar ubiquitin (green), 20-fold molar excess EDGEGEE (purple), or both (blue). **e** Expanded regions of overlayed ¹H-¹⁵N HSQC spectra acquired on samples of 0.1 mM ¹⁵N-labeled hRpn13_ΔMTT alone (black) or at 4-fold (left), 2-fold (middle) or equimolar (right) concentration with K48-diUb^EDGEGEE (blue), K48-diUb^ENT (maroon) or K48-diUb^D77 (green). **f** Cartoon of hRpn13 Pru (purple) interaction with a substrate (blue) ubiquitin chain (yellow) promoting its interaction with a disordered peptide sequence in the substrate. Source data for (**a** and **c**) are provided in the Source Data file.

(Supplementary Fig. 12a). Altogether, these data indicate that hRpn2 and EDGEGEE can to some degree interact simultaneously with hRpn13 Pru.

To further investigate whether there is a concentration at which EDGEGEE or ENT can displace hRpn2 from hRpn13, we performed a competitive binding experiment whereby the fluorescent probe TAMRA (5-carboxytetramethylrhodamine) was attached to hRpn2 (940–953) and binding to hRpn13_ΔMTT measured by fluorescence polarization (FP). As expected, FP changes were measured as hRpn13_ΔMTT was titrated with TAMRA-hRpn2, with a half maximal effective concentration (EC$_{50}$) of 43 ± 7 nM (Supplementary Fig. 12b); no changes were observed following addition of EDGEGEE, ENT, or hRpn2 peptides spanning 940–953 or 944–953 to TAMRA-hRpn2 without hRpn13 present (Supplementary Fig. 12c). Addition of these peptides into a mixture of hRpn13_ΔMTT with a fixed concentration of TAMRA-hRpn2 decreased FP (Fig. 6c), demonstrating competition with TAMRA-hRpn2 for hRpn13. The hRpn2 peptides and EDGEGEE decreased the FP to background levels (~ 92 mP) with estimated EC$_{50}$ values of 0.32 ± 0.02 μM for hRpn2 (940–953), 87 ± 6 μM for hRpn2 (944–953), and 637 ± 51 μM for EDGEGEE. ENT demonstrated lower affinity and required a concentration > 3 mM to completely decrease the FP to background levels. Thus, ENT and EDGEGEE can co-occupy a portion of the hRpn2-binding surface with hRpn2; however, at > 0.5 mM concentrations, these peptides can occupy enough of the hRpn2-binding surface to displace hRpn2.

Altogether, these data indicate that Pru is highly adaptive for low-affinity interactions with a diverse array of peptide sequences at its hRpn2-binding site. We propose that this feature likely allows Pru to bind to peptide sequences of ubiquitinated substrates for subsequent hand-off at proteasomes, possibly to the ATPase ring.

**Pru binds ubiquitin while interacting weakly with peptides**
To test further whether hRpn13 can bind to ubiquitin and the low-affinity peptides concurrently, we added equimolar unlabeled ubiquitin to 100 μM ¹⁵N-labeled hRpn13_ΔMTT preincubated with a 20-fold molar excess EDGEGEE peptide and acquired a ¹H-¹⁵N HSQC spectrum. Comparison of this spectrum to that acquired without ubiquitin indicated retention of interaction with the peptide (Supplementary Fig. 12d), as highlighted by overlapping EDGEGEE-specific effects without and with ubiquitin for E111, K113, and M109 (Fig. 6d). Ubiquitin-specific effects are also observed in the spectrum acquired with both EDGEGEE and ubiquitin present, with further shifting in the ternary complex (Supplementary Fig. 12d), including for L73, C80, and G101 (Fig. 6d).

To test further whether peptide binding can influence hRpn13 Pru interaction with ubiquitin, we generated K48-linked diubiquitin in which the distal moiety (linked via G76) is extended by a 6-residue linker sequence of GSGGSG followed by either the ENT (K48-diUb^ENT) or EDGEGEE (K48-diUb^EDGEGEE) sequence. Varying concentrations of K48-diUb^ENT, K48-diUb^EDGEGEE, or K48-diUb^D77 were added to 100 μM ¹⁵N-labeled hRpn13_ΔMTT and effects on hRpn13 monitored by

recording ¹H-¹⁵N HSQC spectra. In all cases, binding to ubiquitin was observed, as indicated by shifting of signals from amino acids at the ubiquitin contact surface, including loss of the unbound F76 signal (Fig. 6e); arginine substitution of F76 abolishes hRpn13 binding to ubiquitin[13]. A bound state hRpn13 signal was only observed for K48-diUb^EDGEGEE (Fig. 6e), indicating it to form a stronger complex with hRpn13 compared to K48-diUb^ENT or K48-diUb^D77. In addition to F76, other signals provide evidence for hRpn13 having a stronger affinity for K48-diUb^ENT and K48-diUb^D77. Specifically, hRpn13 D78, A100, and G101 signals show more pronounced shifting at sub-stoichiometric concentrations of K48-diUb^ENT and K48-diUb^EDGEGEE compared to K48-diUb^D77 (Fig. 6e). Shifting at the peptide binding site was not evident in the spectra however, as illustrated by D117 (Supplementary Fig. 13), suggesting any direct interactions between the fused peptides and hRpn13 to be too dynamic to capture at this site when at equimolar concentration.

Collectively, these data inspire a model whereby Pru binding to the ubiquitin chain of a ubiquitinated substrate promotes its binding to peptide sequences of the substrate and moreover, since altered shifting is observed at the ubiquitin-binding surface, our data suggest that interactions with peptide sequences, such as would be in substrates, can enhance hRpn13 binding to ubiquitin chains (Fig. 6f).

## Discussion
We serendipitously solved the structure of the hRpn13 Pru domain bound to a short peptide sequence at the extreme N-terminal end of the recombinant protein. Native M1 is buried within a Pru pocket formed by P40, S90 and W108 and at the valley of a U-shaped structure with native sequence along one side and non-native amino acids on the other (Fig. 2a). The interaction is stabilized by trapped water molecules and the crystal lattice, which allows for ENT interaction with three hRpn13 Pru molecules (Fig. 2e and Supplementary Fig. 1g). Between ENT and Pru are 16 residues that span S4 – N20, which are not present in the electron density map. Previous NMR experiments indicate this region to be intrinsically disordered[11]. We expect that these residues do not contribute directly to ENT binding to Pru; however, they tether ENT to Pru, and this spatial restriction could promote binding. In the crystal context, these amino acids are invisible, making it impossible to determine whether ENT is tethered to Pru-A.

We postulate that hRpn13 Pru likely has a preference for ubiquitinated substrates with exposed disordered regions, allowing for bidentate binding to both the ubiquitin chain and the substrate. Such disordered regions may form as a consequence of ubiquitin conjugation, which has been found to destabilize certain protein folds[34]. hRpn13 is present both on and off proteasomes in cells[18,33] and the ENT:Pru interaction is centered around W108, P40, and S90, which also play a critical role in hRpn13 interaction with its proteasome-binding site at the C-terminus of hRpn2[22–24]. This overlapping interaction surface, coupled with the strong affinity of hRpn2 for hRpn13[23] suggests that substrate peptides could be displaced from the

proteasome-binding region by hRpn2. Nonetheless, we find evidence in NMR data acquired on a sample of hRpn13 Pru, hRpn2, and EDGEGEE of simultaneous binding to hRpn2 and EDGEGEE (Supplementary Fig. 12a). This finding suggests that the EDGEGEE interaction may occur at the N-terminal end of the Pru helix, near where hRpn2 P942 binds, as supported by this region being most affected following peptide addition (Supplementary Figs. 10d, 10e, and 11c).

Efficient degradation by the 26S proteasome requires an unstructured initiation region in the substrate to engage the ATPase motor of the proteasome RP[35–37]. Handoff of a disordered substrate peptide sequence from hRpn13 Pru to the ATPase ring could allow that substrate region to serve as an initiation sequence for degradation (Fig. 7). This model proposes a more organized substrate handoff from hRpn13 to the proteasome compared to a previous model whereby the substrate is not bound to a proteasome ubiquitin receptor[38].

In addition to inspiring a model of hRpn13 bidentate binding to ubiquitinated substrates, the interactions observed between Pru and ENT involving M1 can be used to aid virtual screens aimed at improving the potency of small-molecule hRpn13 binders. ENT binds Pru at the edge of the interaction surface for hRpn13-targeting compound **XL44**[32], with M1 at the **XL44** benzamide end where they each interact with the indole ring of W108 (Supplementary Fig. 14). By contrast, the analogous 4-methyl benzamide of **XL5** is directed away from W108[28] (Supplementary Fig. 14). The limited spatial overlap with **XL44** and **XL5** suggests that these molecules can be extended by mimicking interactions with ENT and taking better advantage of the M1-binding pocket. The **XL5**-based hRpn13 PROTAC has been improved 2-fold by optimizing the linker region between **XL5** and the E3 ligase warhead[39]. Further optimization of the linker to mimic ENT interactions with the N-terminal end of the hRpn13 helix, as occurs in Pru-B, could aid this effort.

Altogether, we find that hRpn13 Pru binds promiscuously to peptide sequences at low affinity, which may reflect how it interacts with ubiquitinated substrates. This capacity to bind peptides with plasticity may lead to a more organized handoff of ubiquitinated substrates from hRpn13 to substrate processing centers, especially the ATPases, of the 26S/30S proteasome than previously modeled.

## Methods

### Peptide synthesis
The ENT, ENT_M1S, EDGEGEE, and hRpn2 (940–953) were synthesized by GenScript Biotech and sent as lyophilized powder. The peptides were dissolved in either 100 mM NaCl to 10 mM concentration or NMR buffer with 1 mM TCEP to 50 mM concentration. Storage was at −20 °C.

### Expression plasmids
*Escherichia coli* BL21(DE3) pLysS cells (Invitrogen, 440307) were transfected with a recombinant plasmid of hRpn13 Pru (1–150) cloned into the pRSET vector with a histidine tag at the N-terminus, followed by a PreScission protease cleavage site, as previously described[12]. The expression plasmid for hRpn13_ΔMTT was synthesized by GenScript Biotech using the pGEX-4T-1 vector and an in-frame GST tag at the N-terminus, followed by a thrombin cleavage site. The expression sequence was codon optimized for expression in *Escherichia coli*. Expression plasmids for ubiquitin-ENT and ubiquitin-EDGEGEE were synthesized by GenScript Biotech and cloned into the pET-3a vector. These two constructs added to the human ubiquitin C-terminal end an additional 6-residue linker sequence of GSGGSG, followed by either the ENT (K48-diUb[ENT]) or EDGEGEE (K48-diUb[EDGEGEE]) sequence.

### Expression and purification of hRpn13 proteins
hRpn13 (1–150) plasmid was transfected into *Escherichia coli* strain BL21(DE3) pLysS cells (Invitrogen, 440307), the cells grown at 37 °C in Luria-Bertani (LB) medium supplemented with 100 μg/mL ampicillin and 34 μg/mL chloramphenicol to an optical density (OD) at 600 nm of 0.6 − 0.8 units, and protein expression induced by 0.4 mM of iso-propyl-1-thio-β-D-thiogalactopyranoside (IPTG, UBPBio) exposure for 20 h at 17 °C. For the NMR experiments, hRpn13 (1–150) and hRpn13_ΔMTT were expressed in M9 minimal media supplemented with [15]N ammonium chloride, and hRpn13_ΔMTT was expressed in BL21 (DE3) cells (Invitrogen, 440048) grown in M9 supplemented with 100 μg/mL ampicillin.

Cells expressing hRpn13 (1–150) for crystallography were harvested by centrifugation at 5000 x *g* and 4 °C for 40 minutes by using a Beckman Colter J6-M1 centrifuge with a JS-4.2 rotor and subsequently

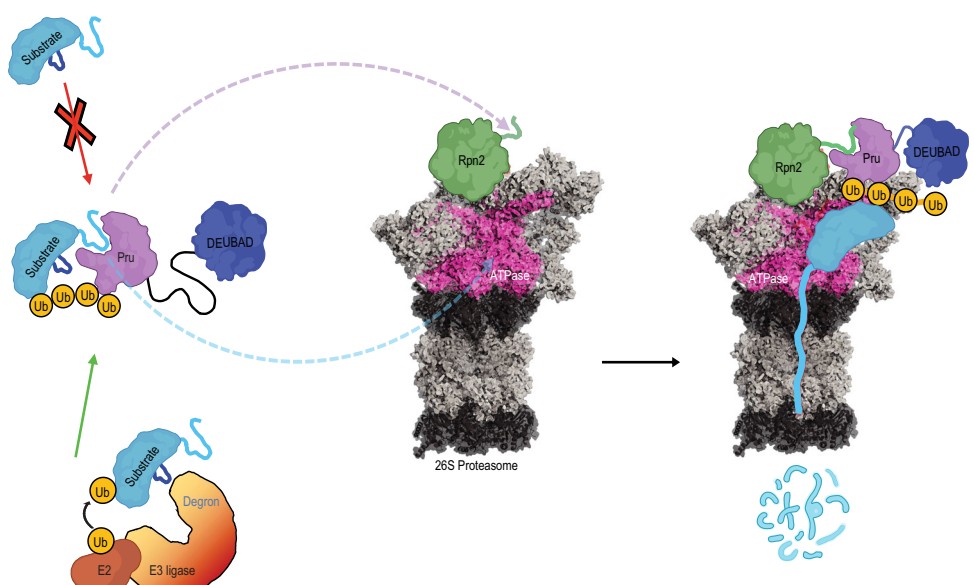

**Fig. 7 | Model of hRpn13 delivery of a ubiquitinated substrate to the 26S proteasome.** hRpn13 Pru binds to hRpn2 (green), which thereby displaces a bound substrate peptide sequence from hRpn13 as it remains bound to the ubiquitin chain. Release from hRpn13 allows the peptide sequence to engage the ATPase (pink) and serve as an initiation sequence for substrate degradation.

**Table 1 | Data collection and refinement statistics (molecular replacement)**

|  | hRpn13:ENT complex |
|---|---|
| **Data collection** |  |
| Space group | P2₁2₁2₁ |
| Cell dimensions |  |
| a, b, c (Å) | 31.26, 56.97, 77.79 |
| α, β, γ (°) | 90, 90, 90 |
| Resolution (Å) | 38.9-1.85 (1.916 – 1.85) [a] |
| $R_{sym}$ or $R_{merge}$ | 0.07717 (0.2239) |
| I / σI | 12.49 (6.97) |
| Completeness (%) | 99.27 (99.83) |
| Redundancy | 6.4 (6.7) |
| **Refinement** |  |
| Resolution (Å) | 1.85 |
| No. reflections | 79094 (8084) |
| $R_{work}$ / $R_{free}$ | 0.1951/0.2160 |
| No. atoms |  |
| Protein | 968 |
| Ligand/ion | 8 |
| Water | 61 |
| B-factors |  |
| Protein | 17.76 |
| Ligand/ion | 26.23 |
| Water | 25.56 |
| R.m.s. deviations |  |
| Bond lengths (Å) | 0.011 |
| Bond angles (°) | 1.31 |

[a]Values in parentheses are for the high-resolution shell.

resuspended in lysis buffer 1 (20 mM sodium phosphate pH 6.5, 300 mM NaCl, and an EDTA-free protease inhibitor cocktail tablet (Roche Diagnostics 11836170001)). The resuspended cells were lysed by sonication followed by centrifugation at 31,000 g for 40 min at 4 °C. The supernatant was then applied to pre-equilibrated Talon Metal Affinity resin (Takara Bio) and incubated for two hours at 4 °C. The resin was extensively washed with buffer A (20 mM sodium phosphate, 300 mM NaCl, pH 6.5) and then buffer B (20 mM sodium phosphate, 50 mM NaCl, pH 6.5). The bound hRpn13 Pru (1–150) proteins were eluted by incubating the resin with 50 units per mL of PreScission protease dissolved in buffer B. Purity of the eluted fractions were analyzed by SDS-PAGE and then loaded onto a Superdex 75 gel filtration column (GE Healthcare) equilibrated with buffer C (20 mM HEPES pH 7.5, 100 mM NaCl and 1 mM TCEP) as the final protein buffer for crystallization. The purification procedure for hRpn13 (1–150) used in NMR experiments was identical, but with the inclusion of 2 mM DTT as a reducing agent at all steps.

For purification of $^{15}$N-labeled hRpn13_ΔMTT, cells were resuspended in lysis buffer 2 (20 mM HEPES at pH 7.5, 300 mM NaCl, 10 mM βME and cOmplete Mini protease inhibitor cocktail tablet (Roche Diagnostics 118361530001)) and lysed by sonication. The cell lysate was then centrifuged at 31,000 × g for 40 min and 4 °C. The supernatant was then applied to the glutathione S-sepharose beads (Cytiva 17-0756-05) pre-equilibrated with the lysis buffer and incubated for three hours at 4 °C. The beads were next washed with wash buffer D (20 mM HEPES at pH 7.5, 300 mM NaCl, 10 mM βME) and then wash buffer C. The protein was eluted with 10 mM reduced glutathione (GSH) in buffer C. The eluted fractions were collected and concentrated. hRpn13_ΔMTT was cleaved from GST by incubating with 100 μL of thrombin solution (1 U/μL) (Sigma-Aldrich) at 4 °C for 4–6 h. Further purification was achieved by size exclusion chromatography

(SEC) using a HiLoad 16/600 Superdex 75 prep grade column (GE Healthcare) in buffer C.

### Expression and purification of ubiquitin proteins
Untagged ubiquitin constructs were expressed in Rosetta 2(DE3) cells (Millipore Sigma, 70954) by 0.4 mM IPTG for 16 h at 18 °C, following growth to an $OD_{600}$ value of 0.6–0.8 in LB media supplemented with 100 μg/mL ampicillin. The cells were lysed in 50 mM sodium acetate (pH 4.5) or 50 mM sodium acetate (pH 4.0) for Ub-EDGEGEE by sonication, and the cellular debris removed by centrifugation at 31,000 g for 30 min. The ubiquitin proteins were then purified from the supernatant by cation exchange on a 5 mL prepacked UNOsphere S (Bio-rad, 12009305) or HiTrap SP (GE Healthcare) column with a 0 – 1.0 M NaCl gradient. Further purification was achieved by SEC on a HiLoad 16/600 Superdex 75 prep grade column (GE Healthcare) in buffer C[40]. Ub-ENT and Ub-EDGEGEE required two further rounds of purification on a Superdex 75 Increase 5/150 column (GE Healthcare). The ubiquitin proteins were stored at − 80 °C and dialyzed overnight prior to NMR experiments into buffer E (20 mM sodium phosphate, pH 6.5, 100 mM NaCl, 1 mM TCEP).

### Production of K48-diUb proteins
K48-diUb samples were produced by using a similar method as previously described[41]. Briefly, Ub-D77 (proximal), Ub-ENT (proximal), Ub-EDGEGEE (proximal) and Ub-K48R (distal) were added at a final concentration of 0.8 mM to a reaction buffer (50 mM Tris pH 8.0, 5 mM MgCl₂, 10 mM creatine phosphate, 0.6 U/mL inorg. PPase, 0.6 U/mL creatine kinase, 2 mM ATP) preincubated with 20 μM E2-25K. 0.1 μM E1 (R&D Systems) was added to this reaction system to initiate diUb synthesis, and it was left for 18 h at 37 °C. The reaction was quenched by dilution with 4 volumes of 50 mM sodium acetate (pH 4.5) and passed over a Mono S 5/50 GL column (GE Healthcare). Further purification was achieved by SEC on a HiLoad 16/600 Superdex 75 prep grade column (GE Healthcare) in buffer C.

### Crystallization of the hRpn13 Pru
Purified hRpn13 (1–150) at 560 μM concentration in buffer C was crystallized by microbatch-under-oil at 4 °C with an equi-volume of 0.1 M sodium acetate, pH 4.6 and 22% PEG3350. Before data collection, crystals were cryo-protected in a reservoir solution containing 25% ethylene glycol.

### Data collection, processing, and structure determination
X-ray diffraction data were collected on the Southeast Regional Collective Access Team (SER-CAT) beamline ID-22 (wavelength 1.0 Å) at the Advanced Photon Source, Argonne National Laboratory, equipped with an EIGER 16 M detector. The X-ray data sets were processed and scaled by SER-CAT auto-processing software, and the structure solved by molecular replacement by using the program MolRep[42] with the crystal structure of the hRpn13 Pru (PDB 5IRS) as the search model. All structural refinement was performed by REFMAC5 (CCP4 package) with default parameters[43]. The Matthews coefficient indicated the presence of one molecule per asymmetric unit. The model building was done manually in Coot[44]. The protein atoms were fit into the electron density contoured at 3.0σ and 1.0σ for the Fo-Fc and 2Fo-Fc maps, respectively. Water molecules were fit into the electron density, and refinement was performed. All data collection, refinement, and validation statistics are detailed in the Table 1; the percentage of favored, allowed, and outlier residues in the Ramachandran plot is 100.0, 0.0 and 0.0, respectively. The structure factors for the final structure have been deposited in the RCSB Protein Data Bank under PDB ID 8VWO.

### MD simulations
Simulations of ENT-bound hRpn13 Pru were done by using the crystal structure as starting coordinates. The MD simulations were performed

**Table 2 | Simulation setup and components**

| Simulation | Box Dimensions | Total # atoms | # of water molecules | type of molecule (# of molecules) |
|---|---|---|---|---|
| Trimer | $a = 85.9$, $b = 85.9$, $c = 85.9$, $\alpha = 60.0$, $\beta = 60.0$, $\gamma = 60.0$ | 44450 | 12853 | Water (12853), salt (81), protein (3), peptide (3) |
| Trimer Extension 1 | | | | |
| Trimer Extension 2 | | | | |
| Trimer Extension 3 | | | | |
| Pru-A | $a = 59.2$, $b = 59.2$, $c = 59.2$, $\alpha = 60.0$, $\beta = 60.0$, $\gamma = 60.0$ | 14643 | 4234 | Water (4234), salt (4), protein (1), peptide (1) |
| Pru-A Extension 1 | | | | |
| Pru-A Extension 2 | | | | |
| Pru-A Extension 3 | | | | |
| Pru-B | $a = 65.5$, $b = 65.5$, $c = 65.5$, $\alpha = 60.0$, $\beta = 60.0$, $\gamma = 60.0$ | 19704 | 5911 | Water (5911), salt (35), protein (1), peptide (1) |
| Pru-B Extension 1 | | | | |
| Pru-B Extension 2 | | | | |
| Pru-B Extension 3 | | | | |

with the Desmond simulation package of Schrödinger LLC (version 2022.4) with protonation states of ionizable sidechains determined with PROPKA. Equilibration and production simulations used the OPLS4 forcefield[45]. Each protein complex was placed in a dodecahedron periodic boundary, solvated with the TIP3P water model and a 10 Å buffer. Chloride and sodium ions were added to neutralize the overall charge of the system at a final ionic strength of 0.150 M. The system was equilibrated in the isochoric/isothermal (NVT) ensemble for 1 ns with position restraints on non-solvent heavy atoms with a force constant of 1000 kcal mol⁻¹ Å⁻² . Two additional simulations were performed of 1 ns duration and in the isobaric/isothermal ensemble (NPT). One simulation utilized position restraints on all non-solvent heavy atoms, and the second simulation maintained position restraints on backbone atoms. Production simulations were performed for a total of 100 ns with a 2 fs timestep in the NPT ensemble at a temperature and pressure of 300 K and 1 atm, respectively. The long-range electrostatic interactions were computed by the particle mesh Ewald method, and short-range electrostatic interactions were calculated with a 9 Å cutoff. Temperature and pressure were maintained with the Nose-Hoover chain and the Martyna-Tobias-Klein (MTK) thermostat and barostat, respectively. The trajectories were saved at 0 and 100 ps intervals for analysis. Interactions were analyzed using the program PyMOL (PyMOL Molecular Graphics System, Schrodinger, LLC, version 3.1.5.1). The system setup for the various simulations is described in Table 2.

The final frames of the initial 100 ns simulations were utilized as a starting point for additional simulations that extended for 200 ns. These simulations were performed in triplicate with differing random seed values. Prior to the calculation of RMSD and RMSF values, rotation and translation of protein atoms were removed through alignment of Cα atoms of successive frames to the first frame. In the case of the trimer complex, successive frames were aligned to the Cα atoms of Pru-A in the first frame. RMSF values were calculated for each of the Cα atoms for the initial 100 ns simulation and each of the 200 ns extended simulations. All calculations, including atom pair distances, Cα RMSD, and Cα RMSF, were calculated with the Python (version 3.8.1) api of the Schrödinger package and plotted with GraphPad Prism (version 10.3.1).

### NMR experiments

All $^1$H-$^{15}$N HSQC experiments were acquired on samples dissolved in NMR buffer (20 mM sodium phosphate, pH 6.5, 50 mM NaCl, 1 mM TCEP, 10% D$_2$O) unless specified, in which case 2 mM DTT replaced TCEP. The samples were purified in DTT-containing buffer and then exchanged into NMR buffer by using Zeba Protein Desalting columns (Thermo Fisher Scientific). The pH was measured and adjusted to 6.5. All NMR experiments were performed at 25 °C on a Bruker Advance

600 MHz spectrometer equipped with a cryogenically cooled probe. The data were processed by NMRpipe[46] (version 11.5) and spectra visualized and analyzed with NMR-FAM-SPARKY[47] (version 3.190) by using previous chemical shift assignments[12,23] and NMRbox[48]. The amide nitrogen and hydrogen chemical shift perturbations (CSP) were mapped for each amino acid according to Eq. 1.

$$CSP = \sqrt{0.2\Delta\delta_N^2 + \Delta\delta_H^2} \qquad (1)$$

$\Delta\delta_H$, change in amide proton value (in parts per million); $\Delta\delta_N$, change in amide nitrogen value (in parts per million).

### Competition assay by fluorescence polarization

The competitive binding experiment was adapted from an earlier study, for which fluorescent labeling of hRpn2 was used to determine its binding affinity for hRpn13[23]. hRpn2 (940-953), hRpn2 (944–953), and TAMRA-labeled hRpn2 (940–953) were purchased from New England Peptide (Gardner, MA). EDGEGEE and ENT were purchased from GenScript (Piscataway, NJ). Solutions (3 μL) containing various concentrations of the peptides were combined with 3 μL of hRpn13_ΔMTT and preincubated in phosphate-buffered saline (PBS, Quality Biological, Gaithersburg, MD) supplemented with 1 mM TCEP (Sigma-Aldrich, St Louis, MO) at room temperature for 5 min. Next, 3 μL of TAMRA-hRpn2 in the same buffer was added to the reaction mixture and incubated at room temperature for 5 min The final concentrations of hRpn13_ΔMTT and TAMRA-hRpn2 were 80 nM and 10 nM, respectively. TAMRA was excited at 540 nm, and the polarized emissions were measured at 590 nm on a TECAN Spark microreader (TECAN, Männedorf, Switzerland). Data were fitted with an [Agonist] vs response (three parameters) function for the hRpn13_ΔMTT titration experiment and an [Inhibitor] vs response (three parameters) function for the peptide competition experiments using GraphPad Prism software.

### Reporting summary

Further information on research design is available in the Nature Portfolio Reporting Summary linked to this article.

## Data availability

The structural coordinates for hRpn13 Pru in complex with ENT have been deposited in the RCSB Protein Data Bank (PDB) under accession code 8VWO. Plasmids for ADRM1_MTT_Del_pGEX-4T-1, pET_3a_Ub_ENT and pET_3a_Ub_EDGEGEE for expression of hRpn13_ΔMTT, Ub-ENT or Ub-EDGEGEE were deposited with Addgene with IDs 237412, 234413 and 237414, respectively. Simulation input files and final coordinates have been included as Supplementary Data files. The raw data files for 26 images are included in the Source Data file. Additional data can be

obtained from the corresponding author upon request. Source data are provided in this paper.

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

## Acknowledgements

This work was supported by the Intramural Research Program of the CCR, NCI, NIH (1 ZIABC011490 to K.J.W. and 1ZIABC011904 to B.A.P.W.). We thank Janusz Koscielniak and Asokan Anbanandam for their maintenance of the NMR spectrometers. We are grateful to Xiuxiu Lu for useful discussions and critical reading of the manuscript. This study utilized computational resources of the NIH HPC Biowulf cluster (http://hpc.nih.gov), NMRbox: National Center for Biomolecular NMR Data Processing and Analysis, a Biomedical Technology Research Resource, which is supported by NIH grant P41GM111135 (NIGMS). X-ray diffraction data were collected at the Southeast Regional Collaborative Access Team (SER-CAT) 22-ID-D (or 22-ID-E) beamline at the Advanced Photon Source, Argonne National Laboratory. SER-CAT is supported by its member institutions, equipment grants (S10_RR25528, S10_RR028976 and S10_OD027000) from the National Institutes of Health, and funding from the Georgia Research Alliance. The content of this publication does not necessarily reflect the views or policies of the Department of Health and Human Services, nor does mention of trade names, commercial products, or organizations imply endorsement by the U.S. Government.

## Author contributions

K.J.W. conceived of and supervised the project. B.H. performed the MD simulations, acquired the NMR experiments and analyzed data. M.C. solved and analyzed the crystal structure, prepared samples and performed NMR experiments. H.N. helped generate protein samples. M.N. performed the fluorescence polarization experiments with supervision from B.A.P.W. All authors contributed to writing the manuscript.

## Funding

## Competing interests

The authors declare no competing interests.

## Additional information

**Supplementary information** The online version contains Supplementary Material available at https://doi.org/10.1038/s41467-025-60843-w.

