## [Transparent Peer Review file · Nature Communications]

An adaptive peptide-binding site in ubiquitin receptor hRpn13 revealed by structural studies

Corresponding Author: Dr Kylie Walters

Version 0:

Reviewer comments:

Reviewer #1

(Remarks to the Author)

Rpn13 engages and recruits ubiquitinated proteins to the 26S proteasome using its PRU domain, which can simultaneously bind ubiquitin chains and the proteasomal subunit Rpn2. Based on a serendipitous discovery during the crystallization of the Rpn13 PRU domain along with a series of NMR studies with the PRU domain and unstructured peptides, the authors put forward an intriguing new model in which the PRU domain participates in a bidentate interaction with ubiquitin chains and a disordered region in a substrate. As the authors point out, the low-affinity interaction between RPN13 PRU and the disordered region of a substrate could speed up the search for an initiation site that is thought to occur by the ATPases of the 26S to commence the degradation process.

The crystal structure and MD simulations support the idea that the binding site occupied by unstructured peptides overlaps with the Rpn2 binding region. However, NMR titration data show significant chemical shifts in residues located in the PRU domain's helix, distal from the Rpn2 binding site. The authors attribute this discrepancy between crystallography/MD and NMR results to an allosteric network involving C88, connecting the helix and the Rpn2 binding site. While the model is intriguing, additional experiments would strengthen the conclusions. Specifically, further validation of the interaction between unstructured peptides and the PRU domain is necessary to substantiate the proposed mechanism. Moreover, providing more evidence for the bidentate interaction between the ubiquitinated substrate and the PRU domain would enhance the manuscript.

Specific suggestions:

1. If the unstructured peptides occupy a similar binding region as the C-terminus of Rpn2, as the crystal structure and MD simulations suggest, the unstructured peptides should lessen the binding affinity of the Rpn2 peptide. This can be tested using a competition binding experiment.
2. The bidentate model can be directly tested by comparing the binding affinities of K48 Ub2 and K48 Ub2 covalently attached to an unstructured peptide. The latter can be generated using acyl hydrazide chemistry recently reported by the Gersch lab (Z. Zhao et al. J. Am. Chem. Soc. 145, 38, 20801-20812).

Reviewer #2

(Remarks to the Author)

The manuscript by Chandravanshi et al describes the serendipitous discovery that the Pru domain of Rpn13 binds peptides with low affinity. Rpn2 displaces these peptides, suggesting an appealing model for the degradation of ubiquitinated proteins by the proteasome: Rpn13 binds both the ubiquitin chain and a disordered region of the ubiquitinated protein, the disordered region is displaced when Rpn13 associates with the proteasome via Rpn2. The disordered region can then be captured by the proteasome to initiate degradation. The authors present a thorough series of biophysical experiments that demonstrate Rpn13 binds peptides of varying sequences concomitantly with ubiquitin and Rpn2 displaces these peptides. The novelty of the observations and the novelty of the model make this work a significant contribution.

This reviewer was confused by the paragraph discussing the E3 ligase structures in the discussion section (p14, line 339). The authors should consider omitting this paragraph- proposing that low affinity interactions can be trapped by crystal

lattices does not need more support- it happens all the time with PEG and other additives.

Typo p14, line 329 "pocked" should be "pocket"

Reviewer #3

(Remarks to the Author)

The authors determined through crystallization that the recombinant hRpn13 protein contains a Pru domain bound to its extreme N-terminus (ENT). They further clarified the molecular mechanism underlying the interaction between hRpn13's Pru domain and the ENT by performing molecular dynamics (MD) simulations. Additionally, they confirmed the interaction between the Pru domain and the peptide sequence in solution using nuclear magnetic resonance (NMR) spectroscopy. These findings suggest a novel mechanism by which hRpn13 interacts with ubiquitinated substrates. However, as a specialist in computational chemistry, I find it challenging to recommend publication until the following issues are addressed.

Major Comments:

1. Regarding MD simulations, could the authors clarify if only a single trial was conducted? Typically, 3–5 separate runs with varying initial velocities are necessary to ensure reproducibility. Given the relatively short simulation duration of 100 nanoseconds, I recommend additional trials—either at least five independent runs at 100 nanoseconds or three extended runs of 200 nanoseconds—to robustly validate the observed interactions.
2. Key molecular dynamics analysis plots are missing. Please provide RMSD (root-mean-square deviation) plots for the main chains of the protein and for the alignment of the ENT within the Pru domain. Additionally, RMSF (root-mean-square fluctuation) plots for relevant residues are necessary for a complete evaluation of the stability and flexibility of the protein regions involved.
3. The manuscript lacks discussion on specific interatomic distances and potential new interactions observed during MD simulations. An analysis of whether new interactions emerged over the course of the simulations, especially as compared to the X-ray structural data, would enhance understanding of the dynamics of the hRpn13-ENT interaction.

Minor Comments:

1. The authors utilized the OPLS4 force field in their simulations; the following reference should be cited for proper attribution:
Lu C, Wu C, Ghoreishi D, Chen W, Wang L, Damm W, Ross GA, Dahlgren MK, Russell E, Von Bargen CD, Abel R, Friesner RA, Harder ED. OPLS4: Improving Force Field Accuracy on Challenging Regimes of Chemical Space. *J Chem Theory Comput.* 2021 Jul 13;17(7):4291-4300. doi: 10.1021/acs.jctc.1c00302.

Version 1:

Reviewer comments:

Reviewer #1

(Remarks to the Author)

The authors have addressed this reviewer's concerns. The addition of competitive binding data along with the NMR data showing enhanced binding of K48-diUb(EDGELEE) really improve the manuscript and provide support for a very intriguing model for a bidentate interaction between Rpn13 and a ubiquitinated substrate. I recommend publication.

Reviewer #3

(Remarks to the Author)

The authors have sincerely addressed the comments regarding the MD simulations by performing multiple independent runs and extending the simulation time to 200 ns. They have also added detailed analyses of RMSD, RMSF, and interatomic interactions, thereby enhancing the credibility of their MD simulations. As a result, the manuscript has been improved in the MD simulation section, and I would recommend it for publication.

REVIEWER COMMENTS

Reviewer #1 (Remarks to the Author):

Rpn13 engages and recruits ubiquitinated proteins to the 26S proteasome using its PRU domain, which can simultaneously bind ubiquitin chains and the proteasomal subunit Rpn2. Based on a serendipitous discovery during the crystallization of the Rpn13 PRU domain along with a series of NMR studies with the PRU domain and unstructured peptides, the authors put forward an intriguing new model in which the PRU domain participates in a bidentate interaction with ubiquitin chains and a disordered region in a substrate. As the authors point out, the low-affinity interaction between RPN13 PRU and the disordered region of a substrate could speed up the search for an initiation site that is thought to occur by the ATPases of the 26S to commence the degradation process.

The crystal structure and MD simulations support the idea that the binding site occupied by unstructured peptides overlaps with the Rpn2 binding region. However, NMR titration data show significant chemical shifts in residues located in the PRU domain's helix, distal from the Rpn2 binding site. The authors attribute this discrepancy between crystallography/MD and NMR results to an allosteric network involving C88, connecting the helix and the Rpn2 binding site. While the model is intriguing, additional experiments would strengthen the conclusions. Specifically, further validation of the interaction between unstructured peptides and the PRU domain is necessary to substantiate the proposed mechanism. Moreover, providing more evidence for the bidentate interaction between the ubiquitinated substrate and the PRU domain would enhance the manuscript.

We thank the Reviewer for their suggestions and were able to support our model by performing the suggested competition assay and monitoring hRpn13 binding to K48 diUb without and with tethering to the ENT or EDGEGEE sequences. Please find these additions described in more detail below.

Specific suggestions:

1. If the unstructured peptides occupy a similar binding region as the C-terminus of Rpn2, as the crystal structure and MD simulations suggest, the unstructured peptides should lessen the binding affinity of the Rpn2 peptide. This can be tested using a competition binding experiment.

As suggested, we performed a competitive binding experiment to obtain the expected result for the ENT and EDGEGEE peptides. A TAMRA fluorescent probe was added to hRpn2 (940-953) and changes in fluorescence polarization measured in a mixture with hRpn13 Pru following addition of ENT, EDGEGEE, or hRpn2 positive control peptides. FP was decreased to background levels with IC_{50} values for EDGEGEE and ENT of $637 \pm 51 \mu\text{M}$ and $> 3 \text{ mM}$, respectively. This new data is included as Fig. 6c, Supplementary Fig. 12b and 12c of the revised manuscript.

2. The bidentate model can be directly tested by comparing the binding affinities of K48 Ub2 and K48 Ub2 covalently attached to an unstructured peptide. The latter can be generated using acyl hydrazide chemistry recently reported by the Gersch lab (Z. Zhao et al. J. Am. Chem. Soc. 145, 38, 20801-20812).

Our collective data indicate that hRpn13 Pru affinity for K48 diUb is substantially stronger than for the peptides, to which it binds dynamically and therefore we expected differences in affinity might be challenging to observe. NMR is sensitive to subtle differences however and we therefore used NMR to test whether ENT and/or EDGEGEE increases binding to K48-linked diubiquitin, as recommended by the Reviewer. We expressed distal ubiquitin in frame with a C-terminal linker (GSGGSG) and ENT (K48-diUb^{ENT}) or EDGEGEE (K48-diUb^{EDGEGEE}) sequence and generated a K48-linked diubiquitin molecule. Comparison of 2D spectra of such samples with K48-diUb^{D77} indicates distinct differences, including better observation of F76 for K48-

diUb^{EDGE_{GEE}}-bound hRpn13 and greater shifting for hRpn13 D78, A100, and G101 following addition of K48-diUb^{EDGE_{GEE}} or K48-diUb^{ENT} compared to K48-diUb^{D77} when hRpn13 is in limiting quantities. We did not observe shifting specific for peptide interaction; however, this is likely due to the transient nature of the interaction. These new findings are included as Fig. 6e and Supplementary Fig. 13.

Reviewer #2 (Remarks to the Author):

The manuscript by Chandravanshi et al describes the serendipitous discovery that the Pru domain of Rpn13 binds peptides with low affinity. Rpn2 displaces these peptides, suggesting an appealing model for the degradation of ubiquitinated proteins by the proteasome: Rpn13 binds both the ubiquitin chain and a disordered region of the ubiquitinated protein, the disordered region is displaced when Rpn13 associates with the proteasome via Rpn2. The disordered region can then be captured by the proteasome to initiate degradation. The authors present a thorough series of biophysical experiments that demonstrate Rpn13 binds peptides of varying sequences concomitantly with ubiquitin and Rpn2 displaces these peptides. The novelty of the observations and the novelty of the model make this work a significant contribution.

We thank this Reviewer for their enthusiasm and helpful suggestions.

This reviewer was confused by the paragraph discussing the E3 ligase structures in the discussion section (p14, line 339). The authors should consider omitting this paragraph- proposing that low affinity interactions can be trapped by crystal lattices does not need more support- it happens all the time with PEG and other additives.

We followed the Reviewer's recommendation and deleted this paragraph and the accompanying figure.

Typo p14, line 329 "pocked" should be "pocket"

We have corrected this typo, thanks!

Reviewer #3 (Remarks to the Author):

The authors determined through crystallization that the recombinant hRpn13 protein contains a Pru domain bound to its extreme N-terminus (ENT). They further clarified the molecular mechanism underlying the interaction between hRpn13's Pru domain and the ENT by performing molecular dynamics (MD) simulations. Additionally, they confirmed the interaction between the Pru domain and the peptide sequence in solution using nuclear magnetic resonance (NMR) spectroscopy. These findings suggest a novel mechanism by which hRpn13 interacts with ubiquitinated substrates. However, as a specialist in computational chemistry, I find it challenging to recommend publication until the following issues are addressed.

We have incorporated all suggestions into the revised manuscript and agree that these changes have strengthened our study. Many thanks for these comments.

Major Comments:

1. Regarding MD simulations, could the authors clarify if only a single trial was conducted? Typically, 3–5 separate runs with varying initial velocities are necessary to ensure reproducibility. Given the relatively short simulation duration of 100 nanoseconds, I recommend additional trials—either at least five independent runs at 100 nanoseconds or three extended runs of 200 nanoseconds—to robustly validate the observed interactions.

We have now extended the 100 ns of MD simulations by another 200 ns of three independent runs as recommended. These new data are included and described in the revised manuscript and have yielded new Supplementary Fig. 2-7. We find strong reproducibility for the trimeric Pru:ENT complex whereby ENT remains wedged between Pru-A and Pru-B (Supplementary Fig. 3 and 4). The ENT complexes with only Pru-A or Pru-B demonstrate large RMSD values as ENT loses its U-shape (as was found in the 100 ns simulation). The results of the extended 200 ns runs are similar for two of the three independent runs and show that ENT generally retains its position at the binding sites but with large fluctuations (Supplementary Fig. 6 and 7). In one of the simulations, ENT moves out of the binding site when only Pru-A or Pru-B is present.

2. Key molecular dynamics analysis plots are missing. Please provide RMSD (root-mean-square deviation) plots for the main chains of the protein and for the alignment of the ENT within the Pru domain. Additionally, RMSF (root-mean-square fluctuation) plots for relevant residues are necessary for a complete evaluation of the stability and flexibility of the protein regions involved.

We have now included the RMSF and RMSD values as Supplementary Fig. 2 for the initial 100 ns of MD simulations and Supplementary Fig. 4 for the extended 200 ns of MD for the trimeric Pru:ENT complex. These plots validated the conclusion that Pru-C has more variance and provide further support for the stability of the complex. In addition, we included these plots for Pru-A:ENT and Pru-B:ENT complexes over 100 ns of MD (Supplementary Fig. 5) and for an extended 200 ns of MD (Supplementary Fig. 7).

3. The manuscript lacks discussion on specific interatomic distances and potential new interactions observed during MD simulations. An analysis of whether new interactions emerged over the course of the simulations, especially as compared to the X-ray structural data, would enhance understanding of the dynamics of the hRpn13-ENT interaction.

These data are included in Figure 3b-d for the trimeric Pru complex with ENT and in Figure 4 for the Pru-A:ENT and Pru-B:ENT complexes. In addition, we have included atom-to-atom distance information in new Supplementary Figure 3a for the trimeric:ENT complex and new Supplementary Figure 6 for the Pru-A:ENT and Pru-B:ENT complexes. Although new transient interactions were formed, no new stable interactions were generated by the MD. We have now clarified this point in the revised document by adding "The MD simulations did not yield any new stable interactions that persisted throughout the 200 ns in the three runs for the trimeric Pru:ENT complex."

Minor Comments:

1. The authors utilized the OPLS4 force field in their simulations; the following reference should be cited for proper attribution:

Lu C, Wu C, Ghoreishi D, Chen W, Wang L, Damm W, Ross GA, Dahlgren MK, Russell E, Von Bargen CD, Abel R, Friesner RA, Harder ED. OPLS4: Improving Force Field Accuracy on Challenging Regimes of Chemical Space. *J Chem Theory Comput.* 2021 Jul 13;17(7):4291-4300. doi: 10.1021/acs.jctc.1c00302.

We thank the Reviewer for bringing this omission to our attention and have now added this reference.